# Accumulated Workload Differences in Collegiate Women’s Soccer: Starters versus Substitutes

**DOI:** 10.3390/jfmk8020078

**Published:** 2023-06-12

**Authors:** Maxine Furtado Mesa, Jeffrey R. Stout, Michael J. Redd, David H. Fukuda

**Affiliations:** School of Kinesiology and Rehabilitation Sciences, University of Central Florida, Orlando, FL 32816, USA; maxine.furtado@ucf.edu (M.F.M.); jeffrey.stout@ucf.edu (J.R.S.); redd@ucf.edu (M.J.R.)

**Keywords:** technology, athlete monitoring, player tracking, football, non-starters

## Abstract

The purpose of this study was to estimate the workloads accumulated by collegiate female soccer players during a competitive season and to compare the workloads of starters and substitutes. Data from 19 college soccer players (height: 1.58 ± 0.06 m; body mass: 61.57 ± 6.88 kg) were extracted from global positioning system (GPS)/heart rate (HR) monitoring sensors to quantify workload throughout the 2019 competitive season. Total distance, distance covered in four speed zones, accelerations, and time spent in five HR zones were examined as accumulated values for training sessions, matches, and the entire season. Repeated-measures ANOVA and Student’s *t* tests were used to determine the level of differences between starter and substitute workloads. Seasonal accumulated total distance (*p* < 0.001), sprints (≥19.00 km/h; *p* < 0.001), and high-speed distance (≥15.00 km/h; *p* = 0.005) were significantly greater for starters than substitutes. Accumulated training load (*p* = 0.08) and training load per minute played in matches (*p* = 0.08) did not differ between starters and substitutes. Substitutes had similar accumulated workload profiles during training sessions but differed in matches from starters. Coaches and practitioners should pursue strategies to monitor the differences in workload between starters and substitutes.

## 1. Introduction

Athlete monitoring examines the physiological stress placed on the body due to physical activity, also known as training load [1]. Training load can be measured internally, reflected as a psychophysiological response to physical activity, or externally, reflected as the physical work performed by the body [2]. Tracking training load variables through the use of global positioning systems (GPSs) has become a common practice in collegiate soccer [3,4,5], with research being increasingly conducted on female collegiate soccer athletes in recent years [4,6,7,8,9]. Monitoring training load could be helpful in detecting changes in fatigue levels during competitive periods when extensive physical performance evaluations are not practical [10]. In addition to detecting variations in fatigue, monitoring training load may help maximize the physical potential of each athlete through individualized approaches to training and recovery.

Substitutions have an important influence on the tactical considerations of a coach’s game plan. In collegiate soccer, an unlimited number of substitutions can be made in the second half of a match, whereas in the first half, an athlete who is substituted out must wait for the second half to be substituted back in [11]. Previous research on elite male soccer players demonstrated that substitutes showed higher work rates than the players they substituted in a competitive match [12,13]. Although no differences in successful pass percentages were observed between elite male soccer substitutes and starters, substituted players covered more distance at a high intensity (≥19.8 km/h; 12.4 ± 5.3 m/min) than players who participated in the entire match (9.8 ± 3.2 m/min) or were substituted (11.3 ± 3.2 m/min) due to the replacement of fatigued players or the need for tactical disruptions [14,15,16]. Similarly, decreases in exercise intensity were reduced with the incorporation of substitutes during matches [17].

The number of minutes played by each athlete during soccer matches also affects the physiological preparation in both sexes [6,12]. Although there are data available on the demands of elite female soccer [6,8,9,18,19,20,21,22], there is a larger representation of research on the demands of elite and collegiate male soccer players [11,12,16,17,23,24,25,26]. Additionally, research in the elite female soccer population may not be applicable to the collegiate female soccer population because there are differences in the demands of matches between collegiate and elite female soccer [20,21,27,28]. For example, elite female soccer players cover an average of 10 km per match [18,29,30], while female collegiate soccer players cover less than 10 km and at lower speed thresholds (<15.6 km/h) [31]. When accounting for variations in training strain and training monotony, no significant differences were found between elite female soccer starters and substitutes [22]. Furthermore, elite female starters produced higher maximal running velocities and aerobic capacities than their non-starting or substitute teammates [6]. Conversely, no significant differences were observed in sprint time or submaximal exercise tests between starting and substitute collegiate athletes. However, worthwhile differences were observed when the starters achieved faster 30 m sprint times than substitutes [32]. Furthermore, substitute collegiate soccer players engaged in greater distances of moderate-intensity running (12.1–15.5 km/h) in matches than starting collegiate soccer players [31]. Collegiate female soccer players also experience higher training loads and a decrease in power output during the season [33]. Based on the existing literature, there are conflicting results between starting and substitute players regarding in-game performance indicators, thus emphasizing the importance of identifying the underlying factors behind monitoring workloads.

Substitute players sometimes display better technical qualities than the players on the field for the full game or the players who were replaced [34]. It is important that substitute players maintain fitness and skill throughout the season to match the high loads per minute they experience when entering the game at a later point. Due to a short and congested college season, teams must maximize roster availability by maintaining the fitness of all players. Tracking workload can allow coaches to prescribe ‘top-up’ conditioning sessions for players who do not receive enough training stimuli during the week [25]. Therefore, it is important to be aware of the workloads imposed on starters compared to substitutes to monitor and adapt training sessions. This study aimed to estimate the workloads accumulated by collegiate female soccer players during a competitive season and compare the workloads of the starting players with those of the substitutes on a collegiate female soccer team. The authors hypothesize that the workloads accumulated during one competitive season will be higher in starters than substitutes. 

## 2. Materials and Methods

This study was a retrospective observational study conducted on one National Collegiate Athletics Association (NCAA, Indianapolis, IN, USA) Division I women’s soccer team over the course of one competitive season (August 2019–November 2019). Workload data were only provided for on-field training and competitive matches. A total of 1245 match and training player sessions were used to generate the seasonal accumulated data for the 19 collegiate women’s soccer players evaluated for this study. Match data included 311 data files, while training data included 934 data files for starters and substitutes. For this study, seasonal accumulated data were defined as the summation of all on-field training sessions and matches. Thirteen regulation time and five overtime matches were included, with 139 data files from starters and 172 data files from substitutes. Starters averaged 78 ± 13.66 min per game, while substitutes averaged 36 ± 13.92 min per game. Complete data were available for 54 training sessions, averaging 70 ± 25.36 min per training session, with 381 data files belonging to starters and 553 data files for substitutes.

### 2.1. Subjects

Nineteen NCAA Division I college women’s soccer players (age: 20 ± 1.61 years; height: 1.58 ± 0.06 m; body mass: 61.57 ± 6.88 kg) were included in the analysis. To be included in analysis, athletes had to have been medically cleared to participate in the women’s soccer team’s training sessions and competitive matches and remained healthy with no injuries throughout the season analyzed. Athletes also had to comply with wearing the GPS/heart rate monitor, which included wearing the Polar Team Pro electrode strap on the xiphoid process for the entire duration of the activity. To determine the status of the starters (*n* = 8) versus the substitutes (*n* = 11), a limit of >50% of the total match time for the entirety of the season was used based on previous research [8,35]. Status of starters versus substitutes was held constant for match and training data analysis. The study was conducted in accordance with the Declaration of Helsinki and approved by the Institutional Review Board at the University of Central Florida (IRB #2763) on the 22nd of February in 2021. The current study was a retrospective observational study approved by the team’s coaching staff. Informed consent was obtained from all subjects involved in the study.

### 2.2. Procedures

Athletes were assigned individual GPS/heart rate monitors (Polar Team Pro, Polar Electro, Oy, Finland) and chest straps prior to the start of the season as part of team monitoring during all training sessions and matches. The Polar Team Pro sensors have previously been deemed valid and reliable for total distance, low-speed running (0–13.99 km/h), high-speed running (14–19.99 km/h), and very high speed running (>20 km/h) in the outdoor setting [36]. To prevent inter-unit error, athletes wore the same sensor for all training sessions and matches [36,37]. Athletes were given their respective sensors to clip onto a chest strap with electrodes attached once stepping onto the training or match pitch. Sensor placement on the chest strap was located on the xiphoid process and athletes were instructed to ensure the electrodes on the inside of the chest strap made full contact with skin. Sensors would turn on as soon as contact between skin and electrodes on the chest strap was made. Data collection began as soon as field activity was started, including warm-up, and was concluded as soon as the sport coach stopped a training session or the final whistle was blown by the match official. All sessions were recorded live on an iPad (Apple, Cupertino, CA, USA) with the Polar Team Pro app(version 2.0.4). After the activity stopped, the sensors were collected and placed on the dock to be imported into the Polar Team Pro online database. Data were exported from the dashboard to Microsoft Excel spreadsheets (Excel 2019, Microsoft Corporation, Redmond, WA, USA) for analysis.

Specific heart rate zones were used to quantify intensity [8,35] and defined by the default of Polar Team Pro as zone 1 = 50–60%, zone 2 = 60–70%, zone 3 = 70–80%, zone 4 = 80–90%, and zone 5 = 90–100%. Heart rate zones were calculated using the maximal heart rate obtained from a Yo-Yo Intermittent Recovery Test Level 1 (YYIRT) completed at the beginning of the pre-season. Training load, taken directly from Polar Team Pro default, was defined in arbitrary units as the amount of effort put into a session based on intensity and duration. The intensity of a session was determined by proprietary algorithms developed by Polar, including training history, weight, VO2max, sex, age, and heart rate. A count of the frequency of accelerations was quantified into three previously established thresholds [8,35], as follows: low = 0.5–1.99 m/s^2^, moderate = 2.00–2.99 m/s^2^, and high = 3.00–5.00 m/s^2^. Speed zones were separated into four previously established groups with the following thresholds: walk/stand ≤ 6.99 km/h, jog = 7.0–14.99 km/h, run 15.0–18.99 km/h, and sprint ≥ 19.00 km/h [8]. The run and sprint speed zones (≥15.00 km/h) were combined to define the high-speed distance (HSD), as specified in Jagim et al. [8]. 

### 2.3. Statistical Analysis

Normal distribution was established through visual inspection of normal Q-Q plots. Student’s *t* tests were used to examine distance metrics, sprints, and training load differences between starters and substitutes. A two-way repeated-measures analysis (zone × group) repeated-measures analysis of variance was used to examine the movement characteristics between starters and substitutes. Bonferroni post hoc analysis was used to determine where there were differences when significant main effects were identified. To calculate the level of differences in workload, effect sizes were calculated. Effect sizes were interpreted as follows: 0.2 (trivial), 0.2–0.6 (small), 0.7–1.2 (moderate), 1.3–2.0 (large), >2.0 (very large) [38]. Data are presented as mean ± standard deviation. Statistical software (JASP, V.16, Amsterdam, The Netherlands) was used for all analyses. Statistical significance was set a priori at *p* < 0.05.

## 3. Results

Seasonal accumulated total distance was significantly higher for starters (337.76 ± 26.28 km) compared to substitutes (246.37 ± 39.01 km; t[17] = 5.72, *p* < 0.001) and matches (starters: 201.58 ± 19.82 km vs. substitutes: 107.09 ± 40.65 km; t[17] = 6.69, *p* < 0.001), as shown in Figure 1. Training load during matches was significantly greater for starters (4586.00 ± 1488.32 a.u.) compared to substitutes (2501.73 ± 1150.54 a.u.; t[17] = 3.45, *p* = 0.003), as shown in Figure 2. There was no significant difference in average training load per minute played in matches between starters (3.15 ± 1.12 a.u./minute) compared to substitutes (4.87 ± 2.42 a.u./minute; t[17] = −1.86, *p* = 0.08).

Seasonal accumulated sprints covered were significantly greater for starters (8169.63 ± 440.85 sprints) compared to substitutes (5771.55 ± 906.55 sprints; t[17] = 6.88, *p* < 0.001) and matches (starters: 4879.88 ± 485.43 sprints vs. substitutes: 2363.27 ± 1040.96 sprints; t[17] = 7.07, *p* < 0.001), as shown in Figure 3. Seasonal accumulated high-speed distance was significantly higher for starters (36.87 ± 6.57 km) compared to substitutes (25.55 ± 8.10 km; t[17] = 3.23, *p* = 0.005) and matches (starters: 24.70 ± 5.12 km vs. substitutes: 12.83 ± 6.92 km; t[17] = 4.09, *p* < 0.001), as shown in Figure 4. Seasonal accumulated training load did not differ between starters (8130.88 ± 2026.60 a.u.) and substitutes (6261.00 ± 2201.22 a.u.; t[17] = 1.89, *p* = 0.08).

Total distance (starters: 136.17 ± 22.08 km vs. substitutes: 139.28 ± 9.79 km; t[17] = −0.42, *p* = 0.68), training load (starters: 3544.88 ± 812.05 a.u. vs. substitutes: 3759.27 ± 1480.45 a.u.; t[16.06] = 0.40, *p* = 0.69), sprints (starters: 3289.75 ± 551.14 sprints vs. substitutes: 3408.27 ± 292.41 sprints; t[17] = −0.61, *p* = 0.55), and high-speed distance (starters: 12.17 ± 2.47 km vs. substitutes: 12.72 ± 1.68 km; t[11.58] = −0.55, *p* = 0.60) did not differ between starters and substitutes during training sessions throughout the competitive season.

Table 1, Table 2 and Table 3 show the accumulated workloads by zone for matches only, training sessions only, and all sessions, respectively. Speed zones, heart rate zones, and acceleration zone metrics are displayed in accumulated kilometers, minutes, and counts, respectively. A repeated-measures analysis of variance (ANOVA) on speed zones resulted in a significant speed zone × player status interaction in mean differences between groups, F(1.584, 26.924) = 6.203, *p* = 0.01. A post hoc Bonferroni test showed significantly different seasonal accumulated distances between starters and substitutes in speed zone 1 (*p* = 0.002, d = 1.91) and speed zone 2 (*p* < 0.001, d = 1.33). A repeated-measures ANOVA on speed zones during matches resulted in a significant speed zone × player status interaction in mean differences between groups, F(1.282, 21.789) = 7.498, *p* = 0.008. A post hoc Bonferroni test showed significantly different match distance covered between starters and substitutes in speed zone 1 (*p* = 0.001, d = 2.53) and speed zone 2 (*p* < 0.001, d = 1.30). A repeated-measures ANOVA on seasonal accumulated and match acceleration counts in different acceleration zones resulted in a significant speed zone × player status interaction in mean differences between groups, F(1.006, 17.106) = 14.064, *p* = 0.002 and F(1.009, 17.147) = 40.243, *p* < 0.001, respectively. A post hoc Bonferroni test showed significantly different seasonal accumulated counts of high-zone accelerations between starters and substitutes (*p* < 0.001, d = 1.85). A post hoc Bonferroni test showed significantly different counts of high-zone accelerations during matches between starters and substitutes (*p* < 0.001, d = 3.19).

## 4. Discussion

The objective of this study was to estimate the workloads accrued by collegiate women’s soccer players over the course of the competitive season, which included all games and practices. The workloads of the starters and substitutes for the same squad were compared as a secondary goal. A key finding of the study was the discrepancy between starters and substitutes in overall match workloads. Discrepancies between starters and substitutes were expected based on previous studies [8,23].

Comparing work rates and absolute values of starting and substitute soccer players becomes complex, as the definitions and time limits considered to define non-starting and substitution players vary. Varying time limits for playing status separation complicates the comparison of results between studies. To separate playing status for analysis, a few studies set a minimum threshold for minutes played. For example, Carling et al. [12] characterized those who played a minimum of 10 min per game. Similarly, Gai et al. [26] and Hills et al. [25] specified a minimum playing time of five minutes for inclusion in analysis. Gimenez et al. [24] considered substitutes as players who played less than 65 min per match during the regular season. Other studies separated playing status by percentages. For example, Curtis et al. [23] included players as starters if they started in more than 60% of the total matches in the season. The methodology of this current study was based on that of Jagim et al., where substitutes were considered to be those that played less than 50% of the total match time [8]. Lorenzo-Martinez et al. [34] did not specify playing time, but excluded substitutions made in the first half and during stoppage time.

The substitutes in the current study covered significantly lower total distances (31% average difference), high-speed distances (63% average difference) and numbers of sprints (34% average difference) than starters. Percentages of average difference for significant values were obtained through group averages for starters and substitutes. Similar results were observed in collegiate female soccer players playing in the third division, where starters had significantly greater values of total distance, high-speed distance, training load, and number of sprints during matches and for seasonal accumulated values than substitutes; however, no differences were noted in training sessions [8]. In contrast, elite substitute soccer players likely covered more absolute high-intensity running distances at >4.2 ≤ 5 m/s and >5 ≤ 6.9 m/s (30.5% average difference), and had higher player loads (13.9% average difference), which is calculated differently than the currently examined training load, compared to starting players [24]. The contrast in results may be due to the use of two friendly matches in the previous study versus an entire competitive season in the current study. When considering work rate relative to minutes played in professional male soccer during friendly matches, substitute players covered higher total distances (4.6% average difference compared to starting players) [39]. Previous research reports that playing time can potentially influence running performance indices, such as differences in cadence in the game or pacing strategies during the time spent on the field [40].

The current study did not demonstrate significant differences in external load markers such as total and high-speed distance covered, number of sprints, or training load during training sessions between substitutes and starters (Table 3, Figure 1A,B and Figure 2A,B). It is important to note that despite the discrepancy between the results of the current study and those of the existing literature, significant variation exists between substitution rules in professional and American collegiate soccer. Although professional soccer coaches are only allowed a total of five substitutes during the whole game, collegiate soccer coaches are allowed unlimited substitutions in the second half. Because 60% of collegiate men’s soccer substitutions count as re-substitutions, the workload of substitutes is lower than that of starters during matches because of their limited participation [23]. Supporting this, Vescovi and Favero [31] demonstrated that collegiate female soccer substitutes covered shorter distances at moderate (15% vs. 19%) and high intensity (6% vs. 16%) than the starting players in competitive matches. Comparable results were also found in male soccer substitutes, who showed significantly less heart-rate-weighted training impulse, total distance, and acceleration counts than starters during a competitive season [23]. Furthermore, imposing high loads on low-minute players may put those athletes at a higher risk of injury than players with higher minutes [41]. Similar to the increased risk of injury during weeks of highly loaded preseason training sessions [42], consistently exposing substitute players to higher loads during training sessions while they continue to experience lower loads during matches may pose higher risks of injury. Therefore, it may be of interest for teams to track training loads separately for starters and substitutes throughout a season to monitor for discrepancies.

Accelerations account for 7–10% of the total training load during competitive matches [43], and an increase in weekly accelerations can increase fatigue throughout a competitive season [19]. Acceleration counts provide a more comprehensive understanding of the amount of energy expended during a match [44], allowing a more detailed approach to the physical workload experienced by players during activity. However, significant differences were only observed for matches and seasonal accumulated low-zone accelerations, where starters performed more match and seasonal accumulated accelerations (Table 1). To prevent large spikes in workload for substitutes when trying to fill any missed load during matches, coaches may be able to recreate similar acceleration patterns during small-sided games during training [45]. Sport coaches may opt for varied small-sided game dimensions to elicit the preferred adaptations depending on the training day and its proximity to a match day.

There are a few limitations of the current study that should be noted. The current study only examined physical workloads obtained through a GPS/heart rate sensor. Physical characteristics and workloads do not address the complete picture throughout a competitive season, as sport coaches consider tactical and technical skills when deciding the starting roster and substitutions. Future research should include tactical variables such as pass completion and ball possession to further determine if any differences exist between starters and substitutes. Furthermore, the integrity of the dataset was maintained through a smaller sample size due to the exclusion of non-compliant sensor-wearing athletes by thorough data analysis. Of the 30 members on the team, 36% of the data were unusable due to injury, no GPS/heart rate sensor assignment, or non-compliance. One-third of the unusable data was due to non-compliance of sensor wear. Additionally, the current study did not normalize all data for playing time as we looked at values in an accumulated manner. Although previous studies have compared starters’ and substitutes’ values in an absolute manner [8,23,26,46,47,48,49,50], future research may look to replicate this study with external load variables relative to playing time [16,39,51,52,53,54].

Quantifying workloads allows you to see the physical stress that starters and substitutes face throughout a season. The results of this study indicate the differences in workload between starting and substitute soccer players with varying minutes of activity. Coaches and practitioners should strive to implement strategies to monitor the differences in physical workload between starters and substitutes.

## 5. Conclusions

The results of this study show differences in the seasonal accumulated and match workloads between starters and substitutes on a college women’s soccer team. Starters showed significantly higher accumulated total distance, sprints, and high-speed distance throughout a competitive season and significantly higher absolute total distance, sprints, high-speed distance, and training load during competitive matches. Despite workload differences in matches, no significant differences were observed between starters and substitutes during training sessions.

## Figures and Tables

**Figure 1 jfmk-08-00078-f001:**
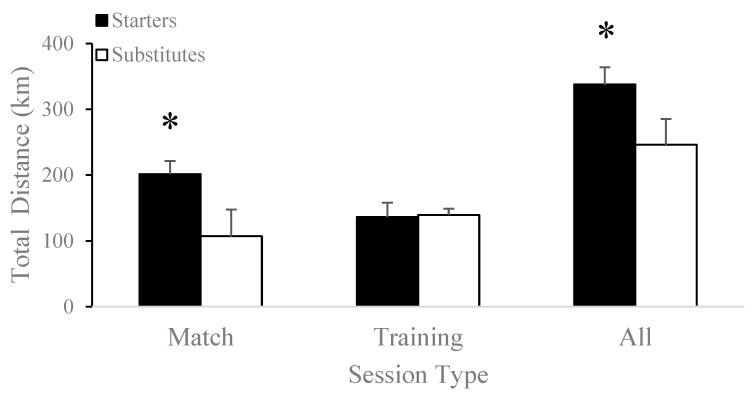
Differences in total distance covered between starters and substitutes in matches, training sessions, and the accumulated competitive season. * *p* < 0.05.

**Figure 2 jfmk-08-00078-f002:**
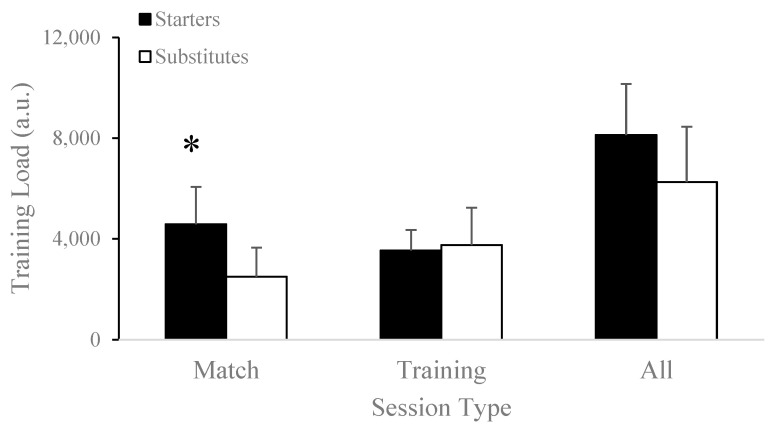
Differences in training load between starters and substitutes in matches, training sessions, and the accumulated competitive season. * *p* < 0.05.

**Figure 3 jfmk-08-00078-f003:**
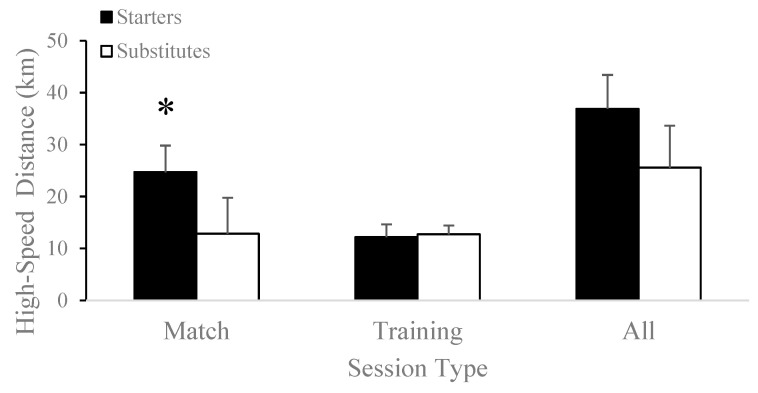
Differences in high-speed distance covered between starters and substitutes in matches, training sessions, and the accumulated competitive season. * *p* < 0.05.

**Figure 4 jfmk-08-00078-f004:**
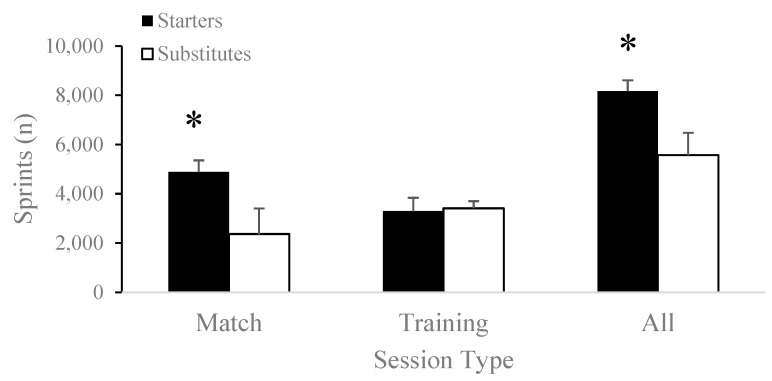
Differences in total count of sprints covered between starters and substitutes in matches, training sessions, and the accumulated competitive season. * *p* < 0.05.

**Table 1 jfmk-08-00078-t001:** Seasonal accumulated workloads for matches and training sessions combined in collegiate Division I women’s soccer players by starting status (mean ± SD).

Variable	Starters	Subs	*p*	Cohen’s d
SP1 (km)	128.20 ± 21.05	94.19 ± 13.94	0.002	1.91
SP2 (km)	149.95 ± 38.52	111.29 ± 13.94	<0.001	1.33
SP3 (km)	22.25 ± 3.28	15.74 ± 4.85	1.000	1.57
SP4 (km)	13.62 ± 4.02	9.81 ± 3.98	1.000	0.95
HR1 (min)	1314.71 ± 451.11	1374.68 ± 401.45	1.000	−0.14
HR2 (min)	1439.75 ± 308.32	1385.06 ± 252.58	1.000	0.19
HR3 (min)	1183.69 ± 204.19	959.23 ± 289.54	1.000	0.90
HR4 (min)	1007.28 ± 420.63	724.15 ± 407.97	1.000	0.68
HR5 (min)	438.42 ± 508.66	262.29 ± 287.59	1.000	0.43
AZ1 (n)	39,203.13 ± 3982.65	31,914.00 ± 3881.33	<0.001	1.85
AZ2 (n)	3480.75 ± 308.05	2530.73 ± 401.91	1.000	2.65
AZ3 (n)	922.25 ± 200.45	654.00 ± 182.24	1.000	1.40

Speed zones: SP1 (walk/stand) ≤ 6.99 km/h; SP2 (jog) = 7.00–14.99 km/h; SP3 (run) = 15.00–18.99 km/h; SP4 (sprint) ≥ 19.00 km/h. Heart rate zones: HR1 = 50–60%; HR2 = 60–70%; HR3 = 70–80%, HR4 = 80–90%; HR5 = 90–100%. Acceleration zones: AZ1 (low) = 0.5–1.99 m/s^2^; AZ2 (moderate) = 2.00–2.99 m/s^2^; AZ3 (high) = 3.00–5.00 m/s^2^.

**Table 2 jfmk-08-00078-t002:** Total accumulated match workloads in collegiate DI women’s soccer players by starting status (mean ± SD).

Variable	Starters	Subs	*p*	Cohen’s d
SP1 (km)	75.39 ± 14.05	40.88 ± 13.20	0.001	2.53
SP2 (km)	101.02 ± 38.72	59.33 ± 23.51	<0.001	1.30
SP3 (km)	16.01 ± 2.63	8.20 ± 4.44	1.000	2.14
SP4 (km)	8.69 ± 2.98	4.63 ± 2.98	1.000	1.36
HR1 (min)	488.28 ± 273.00	581.35 ± 290.02	1.000	−0.33
HR2 (min)	586.96 ± 146.45	514.66 ± 152.88	1.000	0.48
HR3 (min)	584.78 ± 146.26	354.72 ± 112.46	1.000	1.76
HR4 (min)	637.05 ± 251.46	317.62 ± 256.91	0.187	1.26
HR5 (min)	348.33 ± 421.30	112.59 ± 161.85	1.000	0.74
AZ1 (n)	20,021.63 ± 1367.71	12,337.73 ± 3124.69	<0.001	3.19
AZ2 (n)	1901.38 ± 199.73	957.91 ± 400.10	1.000	2.98
AZ3 (n)	456.88 ± 93.15	246.55 ± 116.15	1.000	2.00

Speed zones: SP1 (walk/stand) ≤ 6.99 km/h; SP2 (Jog) = 7.00–14.99 km/h; SP3 (run) = 15.00–18.99 km/h; SP4 (sprint) ≥ 19.00 km/h. Heart rate zones: HR1 = 50–60%; HR2 = 60–70%; HR3 = 70–80%, HR4 = 80–90%; HR5 = 90–100%. Acceleration zones: AZ1 (low) = 0.5–1.99 m/s^2^; AZ2 (moderate) = 2.00–2.99 m/s^2^; AZ3 (high) = 3.00–5.00 m/s^2^.

**Table 3 jfmk-08-00078-t003:** Total accumulated training workloads in collegiate DI women’s soccer players by starting status (mean ± SD).

Variable	Starters	Subs	*p*	Cohen’s d
SP1 (km)	53.51 ± 10.34	53.31 ± 5.88	1.000	0.02
SP2 (km)	48.93 ± 8.20	51.96 ± 5.13	1.000	−0.44
SP3 (km)	7.24 ± 1.39	7.54 ± 0.71	1.000	−0.27
SP4 (km)	4.93 ± 1.31	5.18 ± 1.13	1.000	−0.20
HR1 (min)	826.43 ± 205.33	767.46 ± 215.67	1.000	0.28
HR2 (min)	852.80 ± 188.94	841.87 ± 185.58	1.000	0.06
HR3 (min)	598.91 ± 160.62	599.46 ± 229.99	1.000	−0.002
HR4 (min)	370.23 ± 192.78	399.89 ± 251.78	1.000	−0.13
HR5 (min)	90.09 ± 88.55	138.61 ± 155.54	1.000	−0.38
AZ1 (n)	19,181.50 ± 3746.08	19,576.27 ± 2294.86	1.000	−0.13
AZ2 (n)	1579.38 ± 307.42	1572.82 ± 207.11	1.000	0.003
AZ3 (n)	465.38 ± 118.70	407.46 ± 80.44	1.000	0.57

Speed zones: SP1 (walk/stand) ≤ 6.99 km/h; SP2 (Jog) = 7.00–14.99 km/h; SP3 (run) = 15.00–18.99 km/h; SP4 (sprint) ≥ 19.00 km/h. Heart rate zones: HR1 = 50–60%; HR2 = 60–70%; HR3 = 70–80%, HR4 = 80–90%; HR5 = 90–100%. Acceleration zones: AZ1 (low) = 0.5–1.99 m/s^2^; AZ2 (moderate) = 2.00–2.99 m/s^2^; AZ3 (high) = 3.00–5.00 m/s^2^.

## Data Availability

The data presented in this study are available on request from the corresponding author.

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
