# Peer review of "Accumulated Workload Differences in Collegiate Women’s Soccer: Starters versus Substitutes"

_jfmk, 2023, doi:10.3390/jfmk8020078_

Round 1

Reviewer 1 Report

Introduction

L30 punctuation

academic writing style reviewed throughout to remove 'they' where appropriate

L49 needs re-wording

L55/6 statement needs reference

Add hypothesis

The flow and linking sentences/paragraphs need improving.

Methods

what were player position details? please provide further details regarding starting status - > 50% of the total match time was used - but for how many matches or percentage of total matches?

Detailed description of GPS procedures are needed

The specific heart rate zones used to quantity training intensity - how were these calculated? previous max HR test?

L122 needs clarity of zone speeds and term used to describe? high-intensity, HSR. this needs standardising throughout the paper

were the speed zones based on previous research or the standard threshold within the GPS system? thus details of GPS system are needed

Date for ethics approval needed? was Declaration of Helsinki followed? please state if so

Was informed consent from players gained?

Did the college and head coach approve the study? please detail

Results

L168 - duplicate ==

Discussion

was the key finding expected? - the discrepancy between starters and substitutes in overall match workloads

Standardise format - nonstarters or non-starters? substitutes or non-starters also needs standardising throughout 

L225 needs re-wording 'off of'

L229 - as a key finding - The substitutes in the current study covered significantly lower total distance (31% 229 average difference), high-speed distance (63% average difference) and number of sprints 230 (34% average difference) than starters - needs further explanation 

L248-251 poorly written. needs re-writing 

L251 is referencing format correct?

L244-267 needs re-wording and the key finding made clear and explained in relation to existing literature. Ac/chronic shifts were not examined in the present study but are referenced in this paragraph

L278 the limitations paragraph is poorly written and needs re-writing in an academic style. What would  tactical variables such as pass completion and ball possession add to the paper? please suggest?

match load for starters would clearly be higher than non-starters with no differences in training load - thus, what are the practical implications for this soccer programme?

Reference list does not seem a standardised format. needs checking

Thorough review of paper to ensure paper is written academically. sentence structure in several paragraphs needs improving - highlighted above 

Reviewer 2 Report

Review: Accumulated Workload Differences in Collegiate Women’s Soccer: Starters versus Substitutes

The current investigation assessed the workloads accumulated by collegiate female soccer players during a competitive season and compared the workloads of starters and substitutes.

Starters’ seasonal accumulated total distance (p < .001), sprints (p < .001), and high-speed running distance (p = .005) were greater than substitutes. I would like to commend the authors on the presentation of their manuscript. Overall, it is well written and follows a logical flow from start to finish. The topic is of value to soccer and S&C coaches who are preparing women soccer players to participate at the highest level. I have only a couple of comments that I would like to see addressed. Please highlight these changes by using a different colour text when resubmitting.

Abstract

Comment 1 - Line 11: please insert the year of the season

Comment 2 - L15: please insert the speed threshold used for sprints and high-speed running

Comment 3 - L15: Please update: you have included 4 speed thresholds in your methods. In the abstract, should it be 4 or 5 speed zones? If it is 5, you need to update the methods section.

Introduction

Comment 4 – L30: insert a full stop before Monitoring

Comment 5 – L41-43: you need to clarify the high intensity distances used in this statement. I am sure that those substituted players (those who entered the field) may have covered more high-intensity distance per minute and not an overall distance in metres. As it is written it could be confusing.

Comment 6 – L47-49: can you rewrite this sentence, it does not flow as it is written: Generally, studies have been conducted on elite and collegiate male soccer players [11,12,16–21], although some data on elite female soccer matches and training sessions have been published [6,8,9,22–25].

Comment 7 – L51-52: the use of “ranges” implies that there are two figures, please include a second figure or use “an average of 10 km”

Comment 8 – L53: Please include the speed thresholds or at least less that what speed threshold. For example…. lower speed thresholds (< 17 km.h-1)

Comment 9 – L57: It is unclear what the differences were observed between? Starters had faster 0-30m sprint times compared to non-starters?? Please clarify in the text

Comment 10 – L58-59: please include the speed threshold for “moderate intensity running” and replace incorporated with covered…. substitute collegiate soccer players covered greater moderate intensity (XX km.h-1) running in matches than starting collegiate soccer players

Comment 11 – L65-66: As the sentence is written, it can be interpreted differently. Please update: Substitute players sometimes display better technical qualities than the players on the field for the full game or the players who were replaced.

Comment 12 – L66-68: this sentence is confusing as it appears now. It is important that substitute players maintain fitness and skill throughout the season to match the high loads they experience when entering the game at a later point. Is this “later point” later in the game or if they are required to play a full game later in the competition? If it is later in the game, the high loads they experience when they enter the field are still less that the total volume performed by players who have played the full game. The substitute players may cover high-loads per minute compared to those players who have played the full game. Please update to clarify

Methods

Comment 13 – L79: please insert the year of the season

Comment 14 – L80: please provide a breakdown of the number of matches and training player sessions separately.

Comment 15 – L97: It is unclear from your methods how the subjects were separated into starters and non-starters for training sessions. It is clear for matches. However, a player could be a starter in one game and non-starter in the next game. Can you clarify in your methods how you identified the non-starters in training, was it for the training sessions following the games that they did not start that identified them as non-starters?

Comment 16 – L114: there is overlapping ranges here, for example a heart rate of 60% is placed in which category, in zone 1 or 2? Please update the ranges

Comment 17 – L115-116: It is unclear what scale that training load used? Was it an RPE scale? If so, was is a modified one (0-10). Please provide more details about how you quantified the intensity of the session and how the training load was calculated? Intensity x duration?

Comment 18 – L132: update (Trivial, with (Trivial)

Results

Comment 19 – L137 and 145: Figure 1 does not show where the difference occurs even though it is mentioned in L137. Please update the figure

Comment 20 – L140 and 148: Figure 2 does not show where the difference occurs even though it is mentioned in L140. Please update the figure

Discussion

Comment 21 – L235-238: By stating “non-starting soccer players covered more absolute high-intensity running distances” in your discussion it appears that all distances were higher. The study that you have referenced (19) here showed likely higher differences at speed zones 4.2 m/s and 5 m/s in non-starters than starters. However, the starters showed trivial higher distances at speed zone 6.9 m/s. Please update by changing the sentence or including the speed zones. In addition, the study that you have referenced was conducted using friendly matches and had a total of only two games. This is a major limitation for this study and comparing your finding to this study should be done so with caution. Please highlight this in your discussion as it is difficult to compare with your study which you used competitive games.

Comment 22 – L255: can you define what training load was here?

Comment 23 – The discussion compares the current results with previous findings. However, currently it lacks the possible practical take-aways that coaches could do. For example, line 258-261: Similarly to the increased risk of injury during weeks of highly 258 loaded preseason training sessions [41], consistently exposing substitute players to higher 259 loads during training sessions while they continue to experience lower loads during 260 matches may pose higher risks of injury….. so what should coaches do instead?

Comment 24 – A similar point is made in line 275-277: To prevent large spikes in workload for substitutes when trying to fill any missed load during matches, coaches may be able to recreate similar acceleration patterns during small-sided games during training [45]. The authors should suggest some ways to set up small-sided games to recreate acceleration patterns experienced in the full game.

END

Minor changes to be made. Please see comments for specific comments and line numbers.

Reviewer 3 Report

First of all, I would like to thank the Editor-in-Chief and Associate Editors from Journal of Functional Morphology and Kinesiology for giving me the opportunity to have reviewed the Manuscript ID: jfmk-2394712 titled “Accumulated Workload Differences in Collegiate Women’s Soccer: Starters versus Substitutes”. The main objective of the present study under consideration for publications was to estimate the workloads accumulated by collegiate female soccer players during a competitive season and compare the workloads of the starting players with those of the substitutes on a collegiate female soccer team. No working hypothesis were presented accompanying the study aim. It was also a single-club observational study (in total Nineteen NCAA Division I college women's soccer players participated), that may potentially qualify the design as ‘case study’. In addition, the rationale should be improved, as for example the justification to compare starters and substitutes is not so evident in the introduction. On the other hand, the manuscript text seems well structured. Main conclusions indicated that substitutes have similar accumulated workload profiles during training sessions but differ in matches from starters. As such, I have major revisions that need to be made by the authors in order to attempt polish the manuscript:

P1L23. Introduction should include at least one dedicated paragraph that justify the reason to compare starters and substitutes in the context of the specific population investigated. At present, there is a great amount of information but this essential one is lacking. Please consider also if the following reference can be useful: Fernandes, R.; Brito, J.P.; Vieira, L.H.P.; Martins, A.D.; Clemente, F.M.; Nobari, H.; Reis, V.M.; Oliveira, R. In-Season Internal Load and Wellness Variations in Professional Women Soccer Players: Comparisons between Playing Positions and Status. Int. J. Environ. Res. Public Health 202118, 12817. https://doi.org/10.3390/ijerph182312817

P2L46. “The number of minutes played by each athlete during soccer matches also affects the physiological preparation in both sexes [6,12]. Generally, studies have been conducted on elite and collegiate male soccer players [11,12,16–21], although some data on elite female soccer matches and training sessions have been published [6,8,9,22–25]. However, there are differences in the demands of matches between collegiate and elite female soccer [24–27]…” This whole paragraph needs modification. It starts talking about number of minutes played and then suddenly changed to collegiate versus elite female soccer.

P2L73. “This study aimed to estimate the workloads accumulated by collegiate female soccer players during a competitive season and compare the workloads of the starting players with those of the substitutes on a collegiate female soccer team.” - Please insert study hypothesis

P2L80 and P3L96. “A total of 1,320 match and training player-sessions were used to generate seasonal accumulated data for the 19 collegiate women’s soccer players evaluated for this study.” “Nineteen NCAA Division I college women's soccer players (age: 20 ± 1.61 years; height: 1.58 ± 0.06 m; body mass: 61.57 ± 6.88 kg) were included in the analysis.” “ To determine the status of the starter (n = 8) versus the substitute (n = 11)” - It is necessary to ensure that the number of participants included, and within each group, is sufficient to reach statistical power required to drawn firm conclusions, i.e. a prior sample size estimation and/or inclusion of statistical power for the comparisons made. Also, if any player aged < 18 years-old were used, consider disclose properly on the ethics i.e. whether consent was also obtained from legal guardians

P3L101. “Athletes were assigned individual GPS/ heart rate monitors (Polar Team Pro, Polar Electro, Oy, Finland) prior to the start of the season as part of team monitoring during all training sessions and matches.” - It is mandatory to include information about the validity and reliability of this instrument to collect the dependent variables included in the present work. Please add necessary references and/or supporting data.

P3L125. “Student's t tests were used to examine distance metrics, sprints, and training load differences between starters versus substitutes. A two-way repeated measures analysis (zone x group) repeated measures analysis of variance was used to examine the movement characteristics between starters and substitutes. Bonferroni post hoc analysis was used to determine where there were differences when significant main effects were identified. Normal distribution was established through visual inspection of normal Q-Q plots.” - The verification of normality should be mentioned first in the statistical analysis

P7L210. “The objective of this study was to estimate the workloads accrued by collegiate women’s soccer players over the course of the competitive season, which included all games and practices. The workloads of the starters and substitutes for the same squad were compared as a secondary goal. A key finding of the study was the discrepancy between starters and substitutes in overall match workloads.”  - I appreciate the structure of this paragraph, the only suggestion is to expand the study findings (too vague in the current form). Also, if pertinent include a final sentence indicating the direction of the discussion; this can help clarify some aspects and interest for the readers.

P7L229. “The substitutes in the current study covered significantly lower total distance (31% average difference), high-speed distance (63% average difference) and number of sprints (34% average difference) than starters.”

and

P8L290 “Starters show significantly higher total distance, sprints, and high-speed running distance throughout a competitive season and significantly higher total distance, sprints, high-speed running distance, and training load during competitive matches”

It is of paramount importance to state if this persist also when a normalization to the playing time is take into account given the likely different exposure time according to status of players. Please respond and consider this comment for the remainder variables and it associated discussion/conclusions.

Round 2

Reviewer 1 Report

Unfortunately, regardless of the revised corrections, my initial opinion of this manuscript has not changed. Generally, for publication, the academic writing style needs improving, the limitations need expanding, the conclusions of the paper are not novelty and enhance or improve the existing literature. Furthermore, the numbering format, punctuation (full stops are missing) and spacing needs to be consistent throughout. In summary, the manuscript needs to consider and contribute to the body of knowledge by adding the 'so what' factor. 

Therefore, unfortunately, my recommendation is still reject. 

academic writing style needs improving

Reviewer 2 Report

Thank you for your rely to my comments. I have reviewed your updated article. I have one additional comment that I would like to see addressed.

L41-43 - It would be valuable for the reader to have the mean high-speed distance figures inserted here after the relevant group (substituted, entire match and were substituted) 

Thank you

Reviewer 3 Report

I would like to thank the authors for responding to all my queries. My recommendations is 'minor revisions', because one question still should be treated with more caution:

FIRST REVIEW REPORT

"Point 8: P7L229. “The substitutes in the current study covered significantly lower total distance (31%

average difference), high-speed distance (63% average difference) and number of sprints (34% average

difference) than starters.”  

and  

P8L290 “Starters show significantly higher total distance, sprints, and high-speed running distance

throughout a competitive season and significantly higher total distance, sprints, high-speed running

distance, and training load during competitive matches” 

It is of paramount importance to state if this persist also when a normalization to the playing time is take

into account given the likely different exposure time according to status of players. Please respond and

consider this comment for the remainder variables and it associated discussion/conclusions. 

AUTHORS RESPONSE:

Response 8: Thank you for your point, the current study did not normalize for playing time for all

variables, as this was observing the accumulation of each separate variable. The normalization for playing 

time would have only been possible for matches and not training sessions, as all players trained as a team

for the same duration. "

SECOND REVIEW REPORT
I understand that normalization is possible only for matches, and as a consequence it could be necessary to insert. Otherwise a solid justification should be presented in the paper for not addresing time-differences between starters and reserves, which in turn may potentially lead into misleading conclusions.
